# Cu-based high-entropy two-dimensional oxide as stable and active photothermal catalyst

Yaguang Li [1,2,6] ✉, Xianhua Bai[1,6], Dachao Yuan[2,6], Chenyang Yu[1], Xingyuan San[1], Yunna Guo[3], Liqiang Zhang[3] ✉ & Jinhua Ye [1,4,5] ✉

Cu-based nanocatalysts are the cornerstone of various industrial catalytic processes. Synergistically strengthening the catalytic stability and activity of Cu-based nanocatalysts is an ongoing challenge. Herein, the high-entropy principle is applied to modify the structure of Cu-based nanocatalysts, and a PVP templated method is invented for generally synthesizing six-eleven dissimilar elements as high-entropy two-dimensional (2D) materials. Taking 2D $Cu_2Zn_1Al_{0.5}Ce_5Zr_{0.5}O_x$ as an example, the high-entropy structure not only enhances the sintering resistance from 400 °C to 800 °C but also improves its $CO_2$ hydrogenation activity to a pure CO production rate of 417.2 mmol g$^{-1}$ h$^{-1}$ at 500 °C, 4 times higher than that of reported advanced catalysts. When 2D $Cu_2Zn_1Al_{0.5}Ce_5Zr_{0.5}O_x$ are applied to the photothermal $CO_2$ hydrogenation, it exhibits a record photochemical energy conversion efficiency of 36.2%, with a CO generation rate of 248.5 mmol g$^{-1}$ h$^{-1}$ and 571 L of CO yield under ambient sunlight irradiation. The high-entropy 2D materials provide a new route to simultaneously achieve catalytic stability and activity, greatly expanding the application boundaries of photothermal catalysis.

Nanomaterials, with the merits of high catalytic activity and high atomic utilization, play a crucial role in numerous fields such as materials, energy, and chemistry[1–5]. However, due to the high surface activity, nanomaterials tend to sinter into larger particles at elevated temperatures[6–8], resulting in catalytic deactivation[9,10]. Representatively, Cu-based nanomaterials are the benchmark catalysts of diverse industrial processes, such as methanol steam reforming[11], methanol synthesis[12,13], water gas shift reaction[14], and emerging photothermal catalysis[9,15]. But, the Taman temperature (~400 °C) of Cu-based nanocatalysts is always lower than the operating temperature of industrial processes and photothermal catalysis

(450 °C)[16,17], which shortens the service life of industrial catalytic systems and reduces the solar to chemical energy conversion efficiency. To date, strong metal-support interaction (SMSI) is the predominant approach for enhancing the sintering resistance of nanocatalysts[18,19]. Typically, Sun et al. have reported that the 2D silica supported Cu nanoparticles exhibit a stable $CO_2$ hydrogenation rate of ~60 mmol g$^{-1}$ h$^{-1}$ at 500 °C via SMSI[20]. However, SMSI involves partial or complete encapsulation of Cu-based nanoparticles by heterogeneous materials overlayers[21,22], which blocks the active Cu sites, impedes the transport of reactants and loses catalytic activity[23,24]. Therefore, regulating the structure of Cu-based

[1]Research Center for Solar Driven Carbon Neutrality, Hebei Key Lab of Optic-electronic Information and Materials, The College of Physics Science and Technology, Institute of Life Science and Green Development, Hebei University, Baoding 071002, China. [2]College of Mechanical and Electrical Engineering, Key Laboratory Intelligent Equipment and New Energy Utilization of Livestock and Poultry Breeding, Hebei Agricultural University, Baoding 071001, China. [3]Clean Nano Energy Center, State Key Laboratory of Metastable Materials Science and Technology, Yanshan University, Qinhuangdao 066004, China. [4]International Center for Materials Nanoarchitectonics (WPI-MANA), National Institute for Materials Science (NIMS), 1-1 Namiki, Tsukuba, Ibaraki 305-0044, Japan. [5]Graduate School of Chemical Science and Engineering, Hokkaido University, Sapporo 060-0814, Japan. [6]These authors contributed equally: Yaguang Li, Xianhua Bai, Dachao Yuan. ✉e-mail: liyaguang@hbu.edu.cn; lqzhang@ysu.edu.cn; Jinhua.YE@nims.go.jp

nanocatalysts to obtain high catalytic stability and activity at high temperatures is important for the catalytic science[25–27].

The structural rigidity of materials is proportional to the structural free energy ($\Delta G = \Delta H - T\Delta S$), where $\Delta H$, $\Delta S$ represent the enthalpy change and entropy change, respectively[28,29]. In physical essence, previously reported stabilization strategies primarily focus on enhancing the structural enthalpy ($\Delta H$)[30,31]. Herein, we proposed a high-entropy concept to strengthen the structural rigidity of Cu based nanocatalysts[32], and a PVP templated method could generally and large-scale synthesize high-entropy two-dimensional (2D) materials. Due to the high structural entropy, 2D $Cu_2Zn_1Al_{0.5}Ce_5Zr_{0.5}O_x$ exhibited superior activity and stability for the reverse water gas shift reaction (RWGS, $CO_2 + H_2 \rightarrow CO + H_2O$) under high temperature and $H_2$/air corrosion[33,34]. Consequently, the 2D $Cu_2Zn_1Al_{0.5}Ce_5Zr_{0.5}O_x$ could be extended to photothermal RWGS under harsh conditions, demonstrating unexpected $CO_2$ conversion rate and solar to chemical energy conversion efficiency. In an outdoor photothermal catalysis test, solar-driven RWGS for 7 continuous days was realized by using 2D $Cu_2Zn_1Al_{0.5}Ce_5Zr_{0.5}O_x$. This work offers a new pathway for low-temperature synthesizing high-entropy metal oxide nanocatalysts to realize the synergism of catalytic stability and activity of Cu based nanocatalysts.

## Results

### Low temperature synthesis of high-entropy two-dimensional materials

Several methods have been applied to synthesize high entropy materials[35], such as the carbothermal shock technique and the fast-moving bed pyrolysis technique[14,36]. However, the synthetic accessibility of these methods is limited by high temperature (usually >1000 °C), specialized equipment and tedious procedures[37,38]. Therefore, it is urgent to develop a low temperature and simple method for preparing high entropy nanocatalysts. As illustrated in Fig. 1a, a polyvinylpyrrolidone (PVP) templated method was employed to synthesize high-entropy two-dimensional (2D) materials. High entropy materials generally contain more than 5 kinds of metal elements, and all elements are generally in equal proportion. To verify the universality of this method, 10 kinds of metal ions ($Ce^{3+}$, $Cu^{2+}$, $Mn^{2+}$, $Mg^+$, $Al^{3+}$, $Co^{2+}$, $La^{3+}$, $Zr^{4+}$, $Ca^{2+}$, $Y^{3+}$) with equal atomic proportion were added into this solution. During aging, the PVP was self-assembled into a 2D micelle (Supplementary Fig. 1). The freeze-drying process was applied to obtain solids of 2D PVP micelles loaded with various metal ions. After annealing the precursors in air at 450 °C, the mixed metal ions formed 2D materials. Figure 1b shows that 2D $Ce_1Cu_1Mn_1Mg_1Al_1Co_1La_1Zr_1Ca_1Y_1O_x$ was grown in 2D morphology, and the eleven elements of Ce, Cu, Mn, Mg, Al, Co, La, Zr, Ca, Y, and O were all evenly distributed on the surface of 2D materials, which is the fingerprint feature of high-entropy materials[36,39]. The powder X-ray diffraction (XRD) pattern of 2D $Ce_1Cu_1Mn_1Mg_1Al_1Co_1La_1Zr_1Ca_1Y_1O_x$ showed a single cubic fluorite phase (Supplementary Fig. 2)[40,41], that belongs to a characteristic crystal structure of high-entropy metal oxide[37,42]. This evidence demonstrates that this method successfully synthesized the 2D high-entropy metal oxides. The preparation temperature of this PVP templated method is only 450 °C, significantly lower than the traditional high entropy material preparation methods (usually >1000 °C)[37,38]. Meanwhile, the instruments, chemicals, and steps required for this PVP templated method are simple and inexpensive. Using this method, we also prepared Cu based 2D high-entropy metal oxides. To optimize catalytic performance, the proportion of $Cu^{2+}$, $Zn^{2+}$, $Al^{3+}$, $Ce^{3+}$, $Zr^{4+}$ was 2:1:0.5:5:0.5. Figure 1c presents the typical transmission electron microscopy (TEM) image of the as-prepared sample. It was clearly observed that the sample had a 2D morphology and no heterogeneous nanoparticles were grown on its surface. The corresponding XRD pattern exhibited four peaks at around 29.4°, 33.7°, 48.2°, and 57.2°, which are indexed to the (111),

(200), (220), and (311) crystal planes of the single cubic fluorite phase (Fig. 1d)[40,41]. Atomically level high-angle annular dark-field scanning TEM (HAADF-STEM) revealed an inter-plane spacing measured to be 3.12 Å, representing the (111) planes of face centered cubic (FCC) phase (Fig. 1e)[43,44]. Furthermore, the elemental mapping images demonstrated the homogeneous distribution of Cu, Zn, Al, Ce, Zr, and O over the whole nanosheet (Fig. 1f). This sample was named 2D $Cu_2Zn_1Al_{0.5}Ce_5Zr_{0.5}O_x$. Atomic force microscopy (AFM) confirmed that the thickness of 2D $Cu_2Zn_1Al_{0.5}Ce_5Zr_{0.5}O_x$ was 4 nm, revealing its ultrathin nature (Supplementary Fig. 3). X-ray photoelectron spectroscopy (XPS) was employed to analyze the elemental chemical states of the prepared sample. The XPS analysis showed that all constituting metal elements of 2D $Cu_2Zn_1Al_{0.5}Ce_5Zr_{0.5}O_x$ were in oxidation states (Supplementary Fig. 4).

### The $CO_2$ hydrogenation activity

Cu-based nanocatalysts are active for reverse water gas shift reaction (RWGS, $CO_2 + H_2 \rightarrow CO + H_2O$), which is a fundamental reaction for the synthesis of methanol (CAMERR process) and alkanes (Fischer-Tropsch processes)[45,46]. 2D $Cu_2Zn_1Al_{0.5}Ce_5Zr_{0.5}O_x$ was applied for RWGS, commercial $CuZnAlO_x$ catalyst (Supplementary Fig. 5, named as $Cu_6Zn_3Al_1$) and Cu doped $CeO_2$ nanosheets (named as 2D $Cu_2Ce_7O_x$, Supplementary Figs. 6–8) were selected as reference samples. The 2D $Cu_2Ce_7O_x$ had the SMSI effect of stabilizing the high dispersion of Cu species[47,48]. Figure 2a shows the RWGS CO production rates of 2D $Cu_2Zn_1Al_{0.5}Ce_5Zr_{0.5}O_x$, 2D $Cu_2Ce_7O_x$ and $Cu_6Zn_3Al_1$ at different temperatures. 2D $Cu_2Ce_7O_x$ and $Cu_6Zn_3Al_1$ showed peak CO generation rates of 50.1 mmol $g^{-1}$ $h^{-1}$ and 35.8 mmol $g^{-1}$ $h^{-1}$ at 450 °C and 400 °C, respectively. Then, the CO generation rates of 2D $Cu_2Ce_7O_x$ and $Cu_6Zn_3Al_1$ slowly dropped along with the further increase operation temperature, indicating their thermal instability. Moreover, the CO production rate of 2D $Cu_2Zn_1Al_{0.5}Ce_5Zr_{0.5}O_x$ was monotonically increased to 417.2 mmol $g^{-1}$ $h^{-1}$ at 500 °C, which was higher than the reported advanced catalysts for RWGS at 500 °C as far as we know (Fig. 2b). For example, $Cu/CeO_{2-\delta}$ (106.2 mmol $g^1$ $h^{-1}$)[49], FeCu/CeAl (102.9 mmol $g^{-1}$ $h^{-1}$)[50], $Pd/TiO_2$ (80 mmol $g^{-1}$ $h^{-1}$)[51], Cu/2D silica (60 mmol $g^{-1}$ $h^{-1}$)[20], $Pt/CeO_2$ (45 mmol $g^{-1}$ $h^{-1}$)[52], Co-Fe/$Al_2O_3$ (18 mmol $g^{-1}$ $h^{-1}$)[53]. Figure 2c displays the thermal RWGS stability of 2D $Cu_2Zn_1Al_{0.5}Ce_5Zr_{0.5}O_x$ at 450 °C for 72 h. The CO production rate of 2D $Cu_2Ce_7O_x$ was reduced from 50 mmol $g^{-1}$ $h^{-1}$ to ~25 mmol $g^{-1}$ $h^{-1}$ after 72 h test, corresponding to 50% inactivation, and the CO production rate of $Cu_6Zn_3Al_1$ was reduced from 36 mmol $g^{-1}$ $h^{-1}$ to 7.2 mmol $g^{-1}$ $h^{-1}$ after 72 h test, corresponding to 80% inactivation. Meanwhile, the CO production rate of 2D $Cu_2Zn_1Al_{0.5}Ce_5Zr_{0.5}O_x$ was maintained at ~355 mmol $g^{-1}$ $h^{-1}$ for 72 h, confirming its thermal stability. Additionally, the 2D $Cu_2Zn_1Al_{0.5}Ce_5Zr_{0.5}O_x$ showed 100% selectivity for $CO_2$ converted as CO (Fig. 2d).

### In-situ characterizations

To directly observe the structure evolution of catalysts during RWGS, in-situ characterization was carried out by using an environmental transmission electron microscope (ETEM) setup, in which $CO_2 + H_2$ acted as feeding gas and the catalysts were heated by a chip to simulate RWGS. The RWGS was carried out on the pristine 2D $Cu_2Zn_1Al_{0.5}Ce_5Zr_{0.5}O_x$ and resulted in no sintering phenomenon during the heating ramp up from 400 °C to 800 °C (Fig. 3a). XRD pattern, HAADF-STEM images and electron diffraction pattern revealed that the crystal structure of 2D $Cu_2Zn_1Al_{0.5}Ce_5Zr_{0.5}O_x$ was also robust after experiencing 800 °C of RWGS (Supplementary Figs. 9–11). In comparison, it was observed that several nanoparticles were sintered on the surface of 2D $Cu_2Ce_7O_x$ during the heating process of RWGS from 400 °C to 800 °C (Fig. 3b). The high-resolution (HR)TEM image confirmed that these nanoparticles were metallic Cu (Supplementary Fig. 12). The $H_2$ temperature-programmed reduction ($H_2$-TPR) was applied to detect their evolution under reduction atmosphere (Fig. 3c).

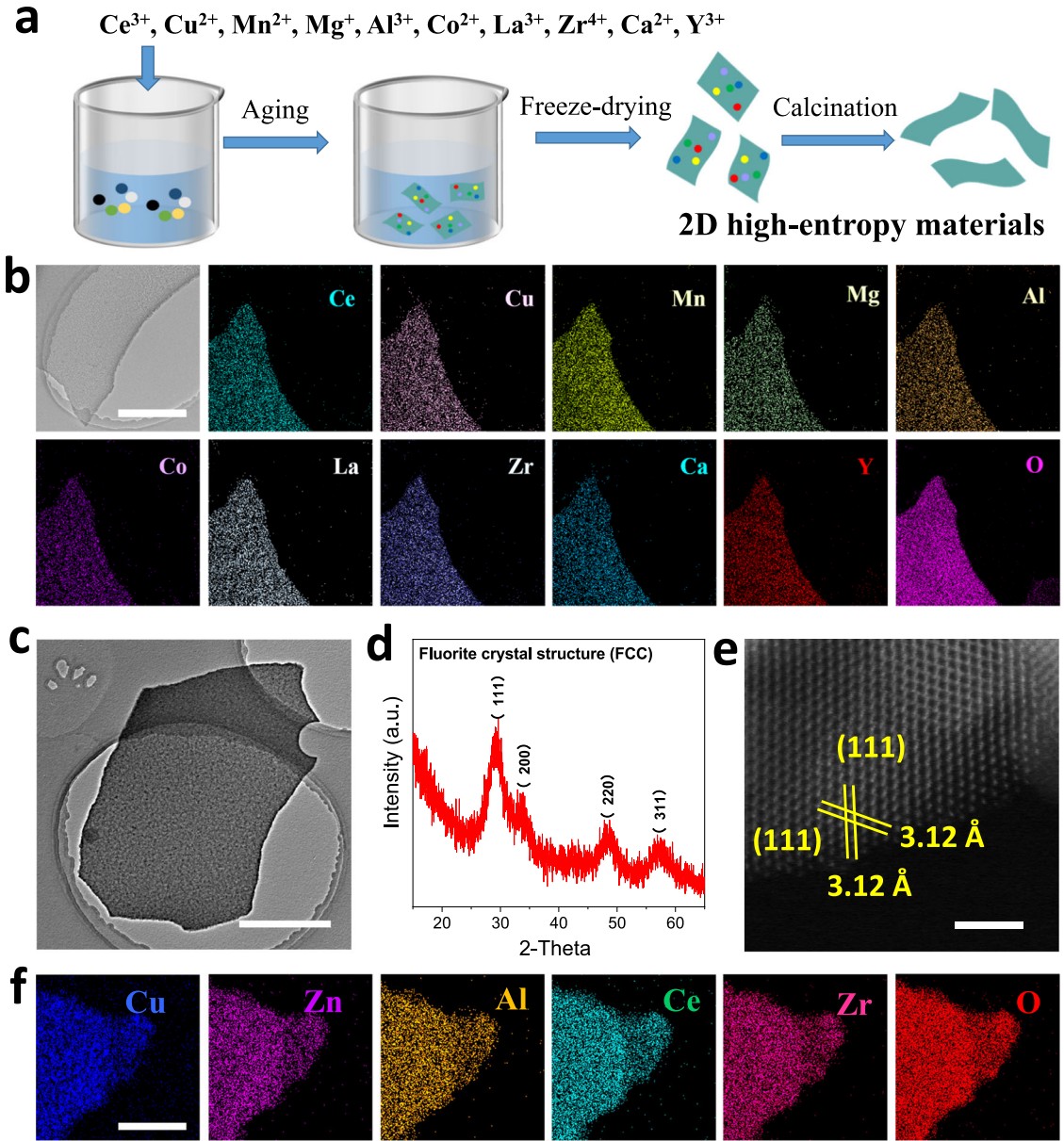

**Fig. 1 | The preparation and characterizations of high-entropy 2D materials.**
**a** The preparation diagram of 2D high-entropy materials. **b** The TEM image and Ce, Cu, Mn, Mg, Al, Co, La, Zr, Ca, Y, O elemental mapping images of 2D $Ce_1Cu_1Mn_1Mg_1Al_1Co_1La_1Zr_1Ca_1Y_1O_x$. **c** TEM image, **d** XRD pattern, **e** HAADF-STEM image of 2D $Cu_2Zn_1Al_{0.5}Ce_5Zr_{0.5}O_x$. **f** Cu, Zn, Al, Ce, Zr, O elemental mapping images of 2D $Cu_2Zn_1Al_{0.5}Ce_5Zr_{0.5}O_x$. The scale bars in **b**, **c**, **e**, **f** are 300 nm, 2 µm, 2 nm, 50 nm, respectively.

The $H_2$-TPR curve of 2D $Cu_2Zn_1Al_{0.5}Ce_5Zr_{0.5}O_x$ had no obvious fluctuations throughout the whole 100–500 °C temperature range (Fig. 3c), revealing the chemical stability of 2D $Cu_2Zn_1Al_{0.5}Ce_5Zr_{0.5}O_x$ under $H_2$ corrosion. While the $H_2$-TPR curve of 2D $Cu_2Ce_7O_x$ showed a reduction peak in the range of 220–300 °C (Fig. 3c), indicating that the Ce and O in 2D $Cu_2Ce_7O_x$ had no obvious valence change (Supplementary Fig. 13), this $H_2$-TPR peak indicated that Cu species undergo a $Cu^{2+}$-$Cu^0$ conversion during the RWGS process[54]. Then, the $H_2$ reduced samples experienced an oxidation process by annealing in air at 300 °C. XPS was used to characterize the chemical states of Cu element before and after $O_2$ corrosion. The XPS spectra of 2D $Cu_2Zn_1Al_{0.5}Ce_5Zr_{0.5}O_x$ shown in Fig. 3d illustrated a similar oxidation state of Cu before and after $O_2$ corrosion. Further, the XPS spectra shown in Fig. 3e confirmed that the Cu species in 2D $Cu_2Ce_7O_x$ was changed from $Cu^0$ to $Cu^{2+}$[55]. It proved that the high-entropy 2D $Cu_2Zn_1Al_{0.5}Ce_5Zr_{0.5}O_x$ had an ultra-stable chemical state under the corrosion of $H_2$ and air. In addition, 2D $Cu_2Zn_1Al_{0.5}Ce_5Zr_{0.5}O_x$ also

showed the activity of CO oxidation (Supplementary Fig. 14)[56], indicating the potential for catalytic versatility.

## Theoretical calculations

Density functional theory (DFT) was applied to investigate the mechanism of sintering resistance and $CO_2$ hydrogenation activity of 2D $Cu_2Zn_1Al_{0.5}Ce_5Zr_{0.5}O_x$. Figure 4a illustrates the atomic structures and metallic Cu precipitation energy barriers of 2D $Cu_2Ce_7O_x$ and 2D $Cu_2Zn_1Al_{0.5}Ce_5Zr_{0.5}O_x$. The metallic Cu precipitation energy barrier of 2D $Cu_2Ce_7O_x$ was 6.61 eV, which was significantly greater than the metallic Cu precipitation energy of pure CuO (1.69 eV, Supplementary Fig. 15). It was confirmed that using metal oxides such as $CeO_2$ as support to introduce SMSI can weaken the sintering of Cu species. Meanwhile, the metallic Cu precipitation energy barrier in 2D $Cu_2Zn_1Al_{0.5}Ce_5Zr_{0.5}O_x$ was as high as 8.85 eV (Fig. 4a), clearly higher than that of 2D $Cu_2Ce_7O_x$ (6.61 eV) and CuO (1.69 eV). Therefore, the sintering resistance of Cu species in 2D $Cu_2Zn_1Al_{0.5}Ce_5Zr_{0.5}O_x$ could be

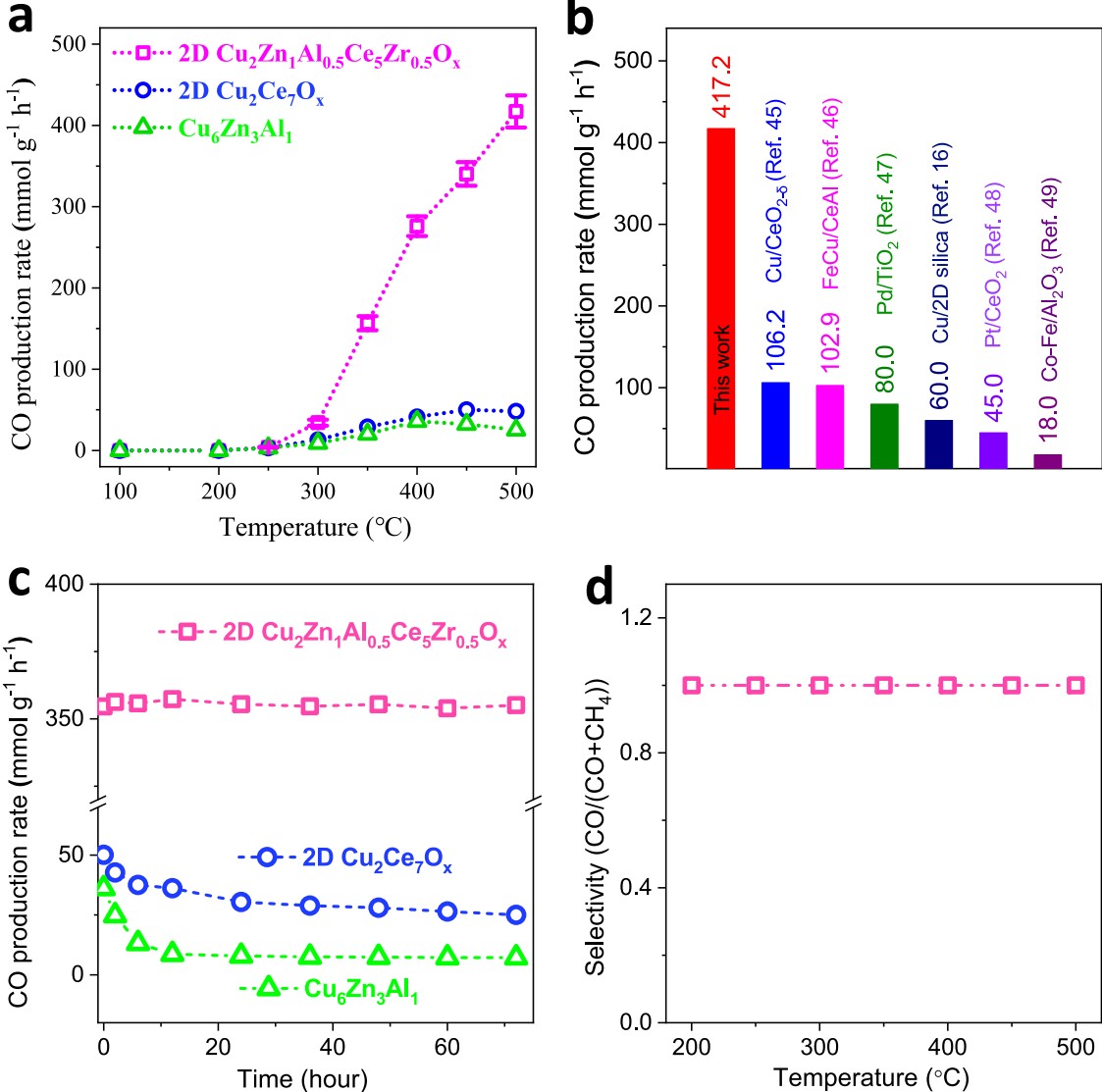

**Fig. 2 | Thermal RWGS performance of catalysts. a** Thermal RWGS performance of 2D $Cu_2Zn_1Al_{0.5}Ce_5Zr_{0.5}O_x$, 2D $Cu_2Ce_7O_x$, commercial $CuZnAlO_x$ ($Cu_6Zn_3Al_1$). **b** Visual contrast diagram of the RWGS CO production rates of 2D $Cu_2Zn_1Al_{0.5}Ce_5Zr_{0.5}O_x$ (This work) and other advanced catalysts at 500 °C. **c** The RWGS stability of 2D $Cu_2Zn_1Al_{0.5}Ce_5Zr_{0.5}O_x$, 2D $Cu_2Ce_7O_x$ and $Cu_6Zn_3Al_1$ under 450 °C. **d** The CO selectivity of 2D $Cu_2Zn_1Al_{0.5}Ce_5Zr_{0.5}O_x$ for thermal RWGS at different temperature. The errors of 2D $Cu_2Zn_1Al_{0.5}Ce_5Zr_{0.5}O_x$ show standard deviation.

mainly attributed to the high-entropy change. Then we simulated the $CO_2$ hydrogenation ($CO_2RR$) of 2D $Cu_2Zn_1Al_{0.5}Ce_5Zr_{0.5}O_x$. Since the in-situ characterization revealed that the Cu species sintered in 2D $Cu_2Ce_7O_x$ during RWGS, the model of 2D $Cu_2Ce_7O_x$ was changed to the atomic structure shown in Fig. 4b, in which the metallic Cu nanoparticle was supported on $CeO_2$ (Cu NP/$CeO_2$). Figure 4c demonstrates the free-energy diagrams and the intermediate pathways of $CO_2RR$ on the Cu NP/$CeO_2$ and 2D $Cu_2Zn_1Al_{0.5}Ce_5Zr_{0.5}O_x$. For the case of Cu NP/$CeO_2$, the release of Cu-CO intermediate ($CO^* + H_2O(g) \rightarrow CO(g) + H_2O(g)$) exhibited a free-energy change of 1.46 eV, marking it as the rate-limiting step. In comparison, the free-energy change of the rate-limiting step of $CO_2RR$ through 2D $Cu_2Zn_1Al_{0.5}Ce_5Zr_{0.5}O_x$ was calculated as 0.74 eV (the formation of Cu-COOH intermediate), which is 0.72 eV lower than that of Cu NP/$CeO_2$ (1.46 eV). In terms of valence electron cloud distribution, the Bader charge of Cu in Cu NP/$CeO_2$, Cu in 2D $Cu_2Zn_1Al_{0.5}Ce_5Zr_{0.5}O_x$, and C in $CO^*$ was calculated as +0.15, +1.38, and −0.22 |e|, respectively. The electronegative value difference revealed that the coordination of Cu-CO in Cu NP/$CeO_2$ and 2D $Cu_2Zn_1Al_{0.5}Ce_5Zr_{0.5}O_x$ were covalent and ionic, respectively[57]. Since

the bond energy of ionic Cu-CO was lower than that of covalent Cu-CO, the dissociation of Cu-CO in 2D $Cu_2Zn_1Al_{0.5}Ce_5Zr_{0.5}O_x$ was easier than that of Cu-CO in Cu NP/$CeO_2$. Therefore, the above results indicated that Cu-CO preferentially dissociation on 2D $Cu_2Zn_1Al_{0.5}Ce_5Zr_{0.5}O_x$ compared to Cu NP/$CeO_2$ due to the transformation of Cu-CO bonding from covalent to ionic.

**Photothermal RWGS**

Photothermal catalysis is a new mode of photochemical energy conversion that can effectively convert solar energy to chemical energy via a pathway involving sunlight-thermal energy-chemical energy, especially under intense sunlight iirradiation[58,59]. This intense irradiation can cause high temperature in photothermal catalysis, thus deactivating nanocatalysts. Due to the excellent sintering resistance, chemical stability and RWGS activity, 2D $Cu_2Zn_1Al_{0.5}Ce_5Zr_{0.5}O_x$ was applied to the photothermal RWGS. The 2D $Cu_2Zn_1Al_{0.5}Ce_5Zr_{0.5}O_x$ was loaded into a homemade TiC-based photothermal device that was designed to fully absorb solar spectrum and convert it to thermal energy, which was used to heat the catalyst (detailed device synthesis

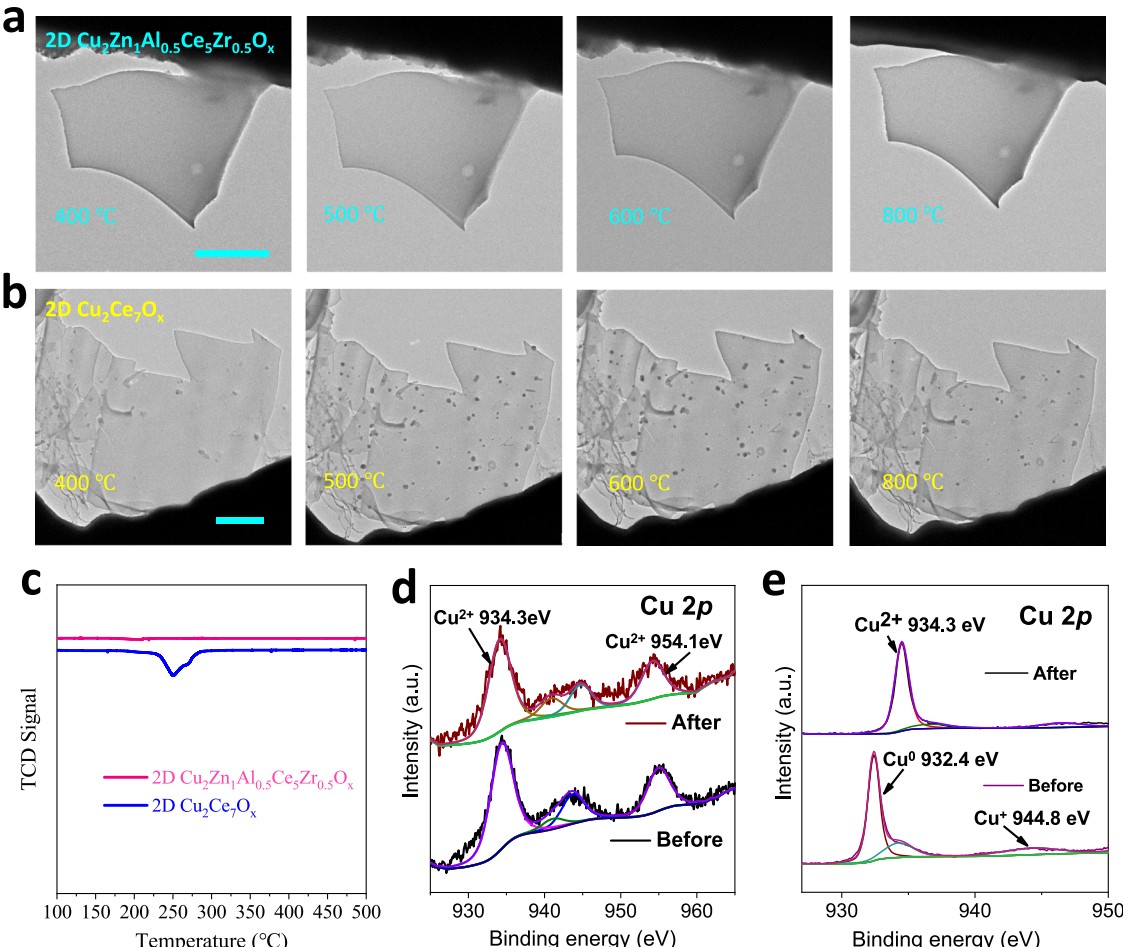

**Fig. 3 | In-situ characterizations of catalysts. a, b** In-situ TEM observations of the 2D $Cu_2Zn_1Al_{0.5}Ce_5Zr_{0.5}O_x$, 2D $Cu_2Ce_7O_x$ at different temperatures of RWGS. **c** $H_2$-TPR curves of the 2D $Cu_2Zn_1Al_{0.5}Ce_5Zr_{0.5}O_x$ and 2D $Cu_2Ce_7O_x$. **d, e** The Cu 2$p$ XPS spectra of 2D $Cu_2Zn_1Al_{0.5}Ce_5Zr_{0.5}O_x$ and 2D $Cu_2Ce_7O_x$ before and after the oxidation process. The scale bars in **a, b** are 1 μm.

can be found in the Supplementary methods and Supplementary Fig. 16). Under 1 sun (1 kW m$^{-2}$ intensity) and 2 suns irradiation, the temperature of the 2D $Cu_2Zn_1Al_{0.5}Ce_5Zr_{0.5}O_x$ catalyst reached 350 °C and 459 °C, respectively (Fig. 5a). The photothermal RWGS measurement revealed that the CO production started at 0.6 sun irradiation, and the CO production rate was 37.4 mmol g$^{-1}$ h$^{-1}$ under 1 sun irradiation.(Fig. 5a). To make a comparison, we listed the state-of-the-art solar driven RWGS in Table 1. Most catalysts irradiated by 1 sun had little RWGS activity and the advanced reported CO generation rates were 0.0013 mmol g$^{-1}$ h$^{-1}$ for $Bi_2In_{2-z}O_{3-x}(OH)_y$[60], 0.0012 mmol g$^{-1}$ h$^{-1}$ for $In_2O_{3-x}(OH)_y$[61]. Therefore, the 1 sun driven photothermal CO generation rate of 2D $Cu_2Zn_1Al_{0.5}Ce_5Zr_{0.5}O_x$ (37.4 mmol g$^{-1}$ h$^{-1}$) was far higher than the previously reported highest value. When the light intensity increased to 2 suns, the CO generation rate of 2D $Cu_2Zn_1Al_{0.5}Ce_5Zr_{0.5}O_x$ was increased to 248.5 mmol g$^{-1}$ h$^{-1}$, at least 31 times higher than the record photothermal RWGS reported under concentrated sunlight (>20 suns) irradiation, e.g., $Bi_xIn_{2-x}O_3$ (8 mmol g$^{-1}$ h$^{-1}$)[62], Pd@HyWO$_{3-x}$ (3 mmol g$^{-1}$ h$^{-1}$)[63], Pd/Nb$_2$O$_5$ (1.8 mmol g$^{-1}$ h$^{-1}$)[64], $In_2O_{3-x}(OH)_y$/Nb$_2$O$_5$ (1.4 mmol g$^{-1}$ h$^{-1}$)[65], Pt/NaTaO$_3$ (0.139 mmol g$^{-1}$ h$^{-1}$)[66], Pd@SiNS (0.01 mmol g$^{-1}$ h$^{-1}$)[67]. The air corrosion photothermal RWGS through 2D $Cu_2Zn_1Al_{0.5}Ce_5Zr_{0.5}O_x$ is shown in Fig. 5b. During the initial 2 suns driven photothermal RWGS, the CO generation rate of 2D $Cu_2Zn_1Al_{0.5}Ce_5Zr_{0.5}O_x$ was remained at ~250 mmol g$^{-1}$ h$^{-1}$ for ~10 h. Then the light and the feeding gas of $CO_2 + H_2$ were cut off for ~5 h. After the light was turned back on, the restarted photothermal RWGS still maintained ~250 mmol g$^{-1}$ h$^{-1}$.

In view of the mass production of 2D $Cu_2Zn_1Al_{0.5}Ce_5Zr_{0.5}O_x$ and the photothermal device, we filled the photothermal device with 100 g 2D $Cu_2Zn_1Al_{0.5}Ce_5Zr_{0.5}O_x$ to make a demonstration to test its application potential (Supplementary Fig. 17). Under 1 sun, 2 suns irradiation, the CO generation rate of this demonstration was 12.3 L h$^{-1}$, 61.5 L h$^{-1}$, respectively (Fig. 5c). According to the experimental data, Fig. 5d showed that the solar to chemical energy conversion efficiency of the demonstration were calculated to be 14.4% and 36.2% under 1- and 2 suns irradiation, respectively (Details seen in Methods). As far as we know, the reported highest solar to chemical efficiency was ~31%[9,68]. This work reveals that the high entropy 2D materials made photothermal catalysis the highest photochemical energy conversion mode in the world. The demonstration was used for industrial outdoor photothermal RWGS. Figure 5e depicts the photograph of an outdoor photothermal RWGS demonstration, in which a TiC/Cu-based device was equipped with a parabolic reflector to concentrate the sparse outdoor sunlight to maintain a high solar driven temperature all day. The photothermal RWGS was tested on 7 successive sunny days in December 2021 in Baoding City of Hebei Province, China. In this continuous outdoor photothermal RWGS, the working time was from 9:00 AM to 16:00 PM, and the rest was the air corrosion time without the supply of feeding gas ($CO_2 + H_2$). As shown in Fig. 5f, the CO yield was 77.6, 62.6, 46.8, 98.2, 88.1, 118.8, 78.7 L on December 12, December 13, December 14, December 17, December 18, December 20, December 21, respectively. And the CO yield difference of each day was originated from the change of sunshine and solar irradiated temperature of the

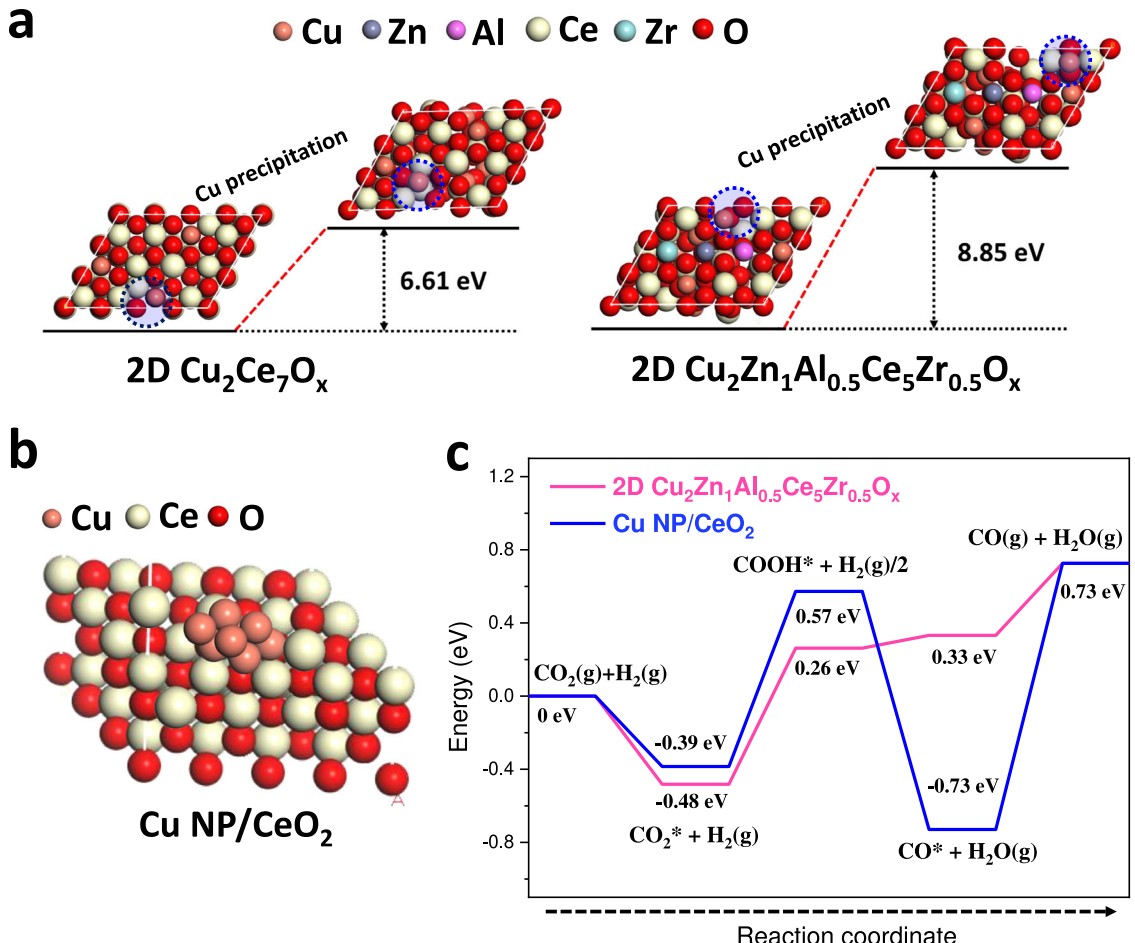

**Fig. 4 | Theoretical calculation of Cu precipitation and CO₂RR. a** Atomic structures of 2D $Cu_2Ce_7O_x$ and 2D $Cu_2Zn_1Al_{0.5}Ce_5Zr_{0.5}O_x$ before and after metallic Cu precipitation with corresponding free energy changes. **b** Atomic structure of Cu NP/CeO₂. **c** Energy profiles for CO₂RR on Cu NP/CeO₂ and 2D $Cu_2Zn_1Al_{0.5}Ce_5Zr_{0.5}O_x$. The *X*-axis illustrates the intermediates; the *Y*-axis illustrates the energy levels of each stage.

catalyst (Supplementary Figs. 18 and 19). It revealed that the 2D $Cu_2Zn_1Al_{0.5}Ce_5Zr_{0.5}O_x$ could realize the continuous operation of outdoor photothermal RWGS.

## Discussion

In this work, a PVP assisted templated method was developed to synthesize high-entropy two-dimensional (2D) materials of 2D $Cu_2Zn_1Al_{0.5}Ce_5Zr_{0.5}O_x$, 2D $Ce_1Cu_1Mn_1Mg_1Al_1Co_1La_1Zr_1Ca_1Y_1O_x$, which showed a single cubic fluorite phase, ~4 nm thickness and uniform elemental distribution. The 2D $Cu_2Zn_1Al_{0.5}Ce_5Zr_{0.5}O_x$ for RWGS showed a stable 417.2 mmol g$^{-1}$ h$^{-1}$ of CO production rate at 500 °C and 100% CO selectivity. The in-situ characterizations revealed that the morphology and crystal structure of 2D $Cu_2Zn_1Al_{0.5}Ce_5Zr_{0.5}O_x$ were robust under 800 °C of RWGS, and the chemical state of 2D $Cu_2Zn_1Al_{0.5}Ce_5Zr_{0.5}O_x$ was rigid under H₂ and air corrosion. DFT calculations revealed that the Cu precipitation energy barrier and RWGS reaction energy barrier over 2D $Cu_2Zn_1Al_{0.5}Ce_5Zr_{0.5}O_x$ was 8.85 eV and 0.74 eV, respectively, due to the high-entropy structure. Under 2 suns irradiation, the 2D $Cu_2Zn_1Al_{0.5}Ce_5Zr_{0.5}O_x$ loaded in a TiC-based device showed a 459 °C temperature of the catalyst, a RWGS CO generation rate of 248.5 mmol g$^{-1}$ h$^{-1}$ and 36.2% solar to chemical energy conversion efficiency. Furthermore, this demonstration was used for outdoor photothermal RWGS for continuous 7 days, exhibiting a CO yield was 77.6, 62.6, 46.8, 98.2, 88.1, 118.8, 78.7 L on December 12, December 13, December 14, December 17, December 18, December 20, December 21, 2021, respectively, under severe changes of natural sunlight. This study

indicated that the high-entropy strategy is a new route for designing nanocatalysts with high activity and stability simultaneously, and promote the application of nanocatalysts. In view of the drastic changes in temperature and atmosphere of natural photothermal catalysis, the high-entropy 2D materials may also provide a cornerstone for the industrialization of natural photothermal catalysis.

## Methods

### Chemicals

Cu(NO₃)₂, Al(NO₃)₃·9H₂O, Zr(NO₃)₄·5H₂O, In(NO₃)₃, Mn(NO₃)₂·4H₂O, Ca(NO)₃·4H₂O, Y(NO)₃·6H₂O, Mg(NO)₂·6H₂O were purchased from Macklin Co., Ltd. Ce(NO₃)₃·6H₂O, La(NO₃)₃·6H₂O, Co(NO₃)₂·6H₂O were purchased from Kermel Co., Ltd. Zn(NO₃)₂·6H₂O and PVP K30 were purchased from Fuchen Chemical Co., Ltd. The Cu₆Zn₃Al₁ catalyst was purchased from Sichuan Shutai Chemical Technology Co., Ltd.

**The synthesis of 2D $Cu_2Zn_1Al_{0.5}Ce_5Zr_{0.5}O_x$.** Firstly, 4 g of PVP was dissolved in 20 ml of H₂O. Then the solution was stirred by a magnetic agitator with the addition of 0.462 g of Cu(NO₃)₂, 0.366 g of Zn(NO₃)₂·6H₂O, 0.231 g of Al(NO₃)₃·9H₂O, 2.675 g of Ce(NO₃)₃·6H₂O and 0.264 g of Zr(NO₃)₄·5H₂O, in which the PVP/metal salts weight ratio was 1. After 0.5 h of stirring, the uniform solution was dripped into liquid nitrogen to make it freeze into ice quickly and it was freeze-dried for 48 h to remove H₂O. The dried product was calcined in a muffle furnace at 450 °C (heating rate 1 °C min$^{-1}$) for 6 h, and the obtained was named 2D $Cu_2Zn_1Al_{0.5}Ce_5Zr_{0.5}O_x$.

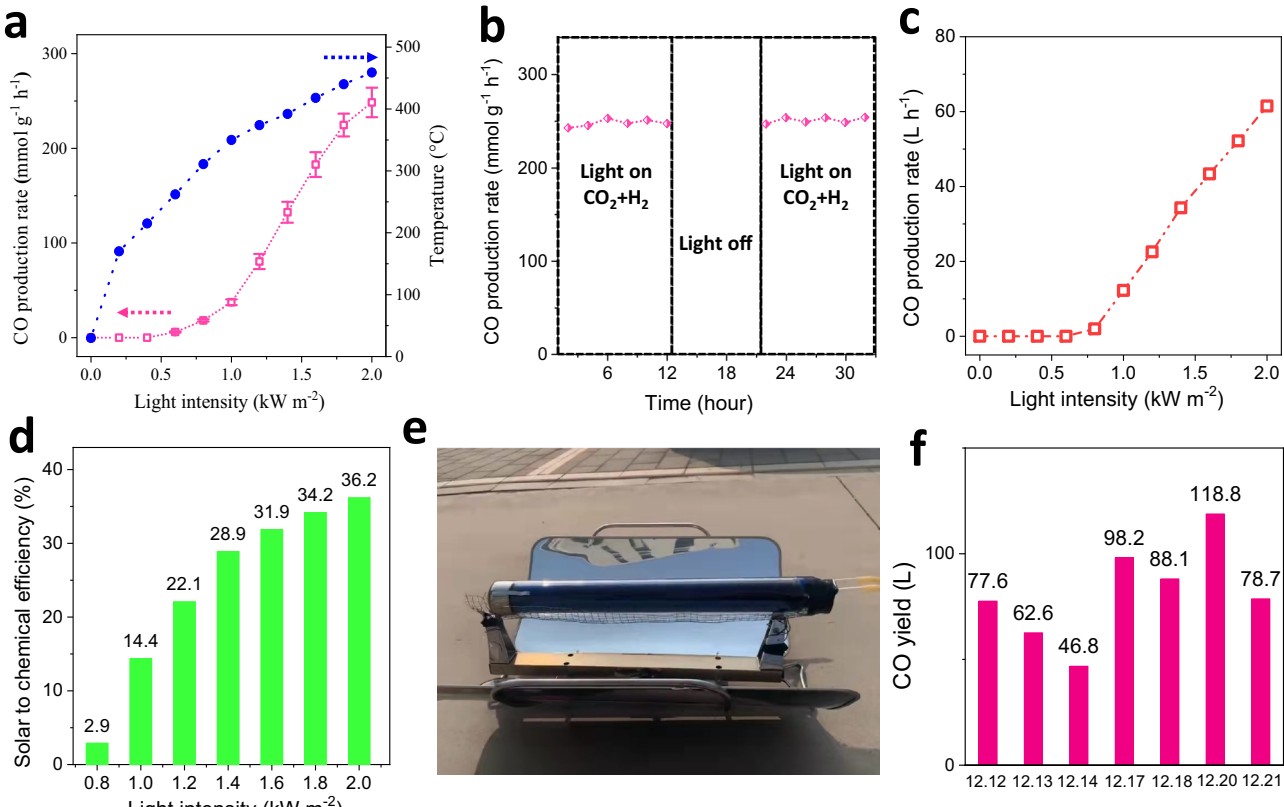

**Fig. 5 | The photothermal RWGS performance of 2D $Cu_2Zn_1Al_{0.5}Ce_5Zr_{0.5}O_x$.**
**a** The temperature of 2D $Cu_2Zn_1Al_{0.5}Ce_5Zr_{0.5}O_x$ and CO production rate of photothermal RWGS through 2D $Cu_2Zn_1Al_{0.5}Ce_5Zr_{0.5}O_x$ under different sunlight irradiation. **b** The CO generation rate of photothermal RWGS through 2D $Cu_2Zn_1Al_{0.5}Ce_5Zr_{0.5}O_x$ under 2 suns irradiation and light off working conditions. **c** The CO generation rate of photothermal RWGS demonstration with 100 g of 2D $Cu_2Zn_1Al_{0.5}Ce_5Zr_{0.5}O_x$ under different sunlight irradiation. **d** The STC efficiency of photothermal RWGS demonstration with 100 g of 2D $Cu_2Zn_1Al_{0.5}Ce_5Zr_{0.5}O_x$ under different sunlight irradiation. **e** The photograph of photothermal RWGS demonstration equipped with reflector in Hebei University. **f** The CO yield of photothermal RWGS demonstration equipped with reflector under outdoor sunlight irradiation, on December 12, 13, 14, 17, 18, 20, 21, 2021, in Baoding City, China. The errors show standard deviation.

**The synthesis of 2D $Ce_1Cu_1Mn_1Mg_1Al_1Co_1La_1Zr_1Ca_1Y_1O_x$.** The synthesis of 2D $Ce_1Cu_1Mn_1Mg_1Al_1Co_1La_1Zr_1Ca_1Y_1O_x$ was similar to the preparation of 2D $Cu_2Zn_1Al_{0.5}Ce_5Zr_{0.5}O_x$, and the only difference was that the metal salts was the mixture of $Ce(NO_3)_3 \cdot 6H_2O$, $Cu(NO_3)_2$, $Mn(NO_3)_2 \cdot 4H_2O$, $Mg(NO)_2 \cdot 6H_2O$, $Al(NO_3)_3 \cdot 9H_2O$, $Co(NO_3)_2 \cdot 6H_2O$, $La(NO_3)_3 \cdot 6H_2O$, $Zr(NO_3)_4 \cdot 5H_2O$, $Ca(NO)_3 \cdot 4H_2O$, $Y(NO)_3 \cdot 6H_2O$ with 1:1:1:1:1:1:1:1:1:1 mole ratio.

**Table 1 | The advanced solar driven RWGS through different catalysts**

| Catalyst | Light intensity | CO rate (mmol· $g^{-1}$· $h^{-1}$) | Refs. |
|---|---|---|---|
| 2D $Cu_2Zn_1Al_{0.5}Ce_5Zr_{0.5}O_x$ | 1 sun | 37.4 | This work |
| 2D $Cu_2Zn_1Al_{0.5}Ce_5Zr_{0.5}O_x$ | 2 suns | 248.5 | This work |
| $Bi_xIn_{2-x}O_3$ | 20 suns | 8 | 62 |
| Pd@HyWO_{3-x} | 20 suns | 3 | 63 |
| $Pd/Nb_2O_5$ | 25 suns | 1.8 | 64 |
| $In_2O_{3-x}(OH)_y/Nb_2O_5$ | none | 1.4 | 65 |
| $Pt/NaTaO_3$ | none | 0.139 | 66 |
| Pd@SiNS | ~15 suns | 0.01 | 67 |
| $Bi_2In_{2-z}O_{3-x}(OH)y$ | 1 sun | 0.0013 | 60 |
| $In_2O_{3-x}(OH)_y$ | 0.8 sun | 0.0012 | 61 |

**The synthesis of 2D $Cu_2Ce_7O_x$.** The synthesis of 2D $Cu_2Ce_7O_x$ was similar to the preparation of 2D $Cu_2Zn_1Al_{0.5}Ce_5Zr_{0.5}O_x$, and the only difference was that the metal salts was the mixture of $Cu(NO_3)_2$, $Ce(NO_3)_3 \cdot 6H_2O$ with 2:7 mole ratio.

**Thermocatalytic RWGS.** The thermocatalytic activity of catalysts for RWGS was tested by the fixed-bed reactor (XM190708-007, DALIAN ZHONGJIARUILIN LIQUID TECHNOLOGY CO., LTD) in continuous flow form. Typically, 20 mg of 2D $Cu_2Zn_1Al_{0.5}Ce_5Zr_{0.5}O_x$ or 200 mg of 2D $Cu_2Ce_7O_x$ or 200 mg of $Cu_6Zn_3Al_1$ catalyst was placed in a quartz flow reactor and the feeding gas of $CO_2/H_2 = 1/1$ with 40 sccm of flow rate was regulated by the mass flow controller. The reaction products were tested by gas chromatography (GC) 7890 A equipped with FID and TCD detectors. Before thermal RWGS, the 2D $Cu_2Ce_7O_x$ and $Cu_6Zn_3Al_1$ with 200 mg weight were reduced by 10% $H_2$/Ar mixture at 300 °C for 4 h with a flow rate of 100 sccm.

**$H_2$-TPR.** Hydrogen temperature-programmed oxidation ($H_2$-TPR) was carried out using an online gas chromatograph (GC-7090A) equipped with a TCD detector. In a typical process, 50 mg of catalyst was placed in a quartz tube (6 mm ID). Subsequently, TPR was performed by heating the samples from room temperature to 500 °C at the heating rate of 5 °C $min^{-1}$, in the presence of a 10% $H_2$/He mixture (50 sccm) flowing.

**Photothermal RWGS.** The photothermal RWGS of 2D $Cu_2Zn_1Al_{0.5}Ce_5Zr_{0.5}O_x$ was similar to Thermocatalytic RWGS over 2D

$Cu_2Zn_1Al_{0.5}Ce_5Zr_{0.5}O_x$ and the difference was that the 2D $Cu_2Zn_1Al_{0.5}Ce_5Zr_{0.5}O_x$ was loaded into a TiC/Cu-based device irradiated by a simulate solar light source (DL3000).

**The photothermal RWGS demonstration.** The photothermal RWGS demonstration was 100 g of 2D $Cu_2Zn_1Al_{0.5}Ce_5Zr_{0.5}O_x$ loaded into a TiC/Cu-based device with 4.4 cm of diameter and 45 cm of length (Supplementary Fig. 16). 120 L h$^{-1}$ of $CO_2$ and 120 L h$^{-1}$ of $H_2$ were simultaneously put into the photothermal RWGS demonstration, which was controlled by a mass flow controller. The photothermal RWGS demonstration was irradiated by a solar light source (DL3000). As shown in Supplementary Fig. 16, the demonstration is irradiated up and down by the light source, so its irradiation area is calculated as 4.4 cm*45 cm*2 = 396 cm$^2$. The composition of produced gas was tested by GC 7890 A equipped with FID and TCD detectors.

**Enthalpy change energy of chemicals.** The enthalpy change energy of $CO_2$ (g), CO (g), $H_2$ (g), $H_2O$ (g) was −393.505, −110.541, 0, −241.818 kJ mol$^{-1}$, respectively. And they are all in gas state.

**Solar to chemical energy conversion efficiency (STC) calculation of photothermal RWGS demonstration.** The STC of photothermal RWGS demonstration was calculated as follows:

$$STC = (\Delta H * \varepsilon / 24.5)/(I * S * 3600) \quad (1)$$

$\Delta H$ was the reaction Enthalpy change energy ($CO_2$ (g) + $H_2$ (g) → CO (g) + $H_2O$ (g), RWGS, $\Delta H = 41.15$ kJ/mol), $\varepsilon$ (L h$^{-1}$) was the CO generation amount per hour detected by a flowmeter, $I$ was the light intensity (kW m$^{-2}$), $S$ was the irradiated area of demonstration (0.0396 m$^2$). The $\varepsilon$ irradiated by 1 sun, 2 suns was 12.3 L h$^{-1}$, 61.5 L h$^{-1}$, respectively, corresponding to 14.4%, 36.2% of STC, severally.

**Outdoor photothermal RWGS.** The photothermal RWGS demonstration was first equipped with a reflector (30 cm width and 55 cm length). 60 L h$^{-1}$ of $CO_2$ and 60 L h$^{-1}$ of $H_2$ were simultaneously put into the demonstration for photothermal RWGS. The composition of produced gas was tested by GC 7890 A equipped with FID and TCD detectors. From 17:00 PM to night to 8:00 AM every day, the supply of $CO_2$ and $H_2$ was stopped from the photothermal RWGS demonstration.

## Data availability
The data generated in this study are provided in the main text and Supplementary information. Extra data are available from the corresponding author upon reasonable request. Source data are provided with this paper.

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

## Acknowledgements

This work is supported by the Natural Science Foundation of Hebei Province (Grant Nos. B2022201090, B2021201074, B2021201034, F2021203097), Hebei Provincial Department of Science and Technology (Grant No. 216Z4303G), Hebei Education Department (Grant Nos. BJ2019016, QN2022059), Interdisciplinary Research Program of Natural Science of Hebei University (Grant Nos. 521100311, DXK202109), the fund of the State Key Laboratory of Catalysis in DICP (Grant No. N-21-05),

the Advanced Talents Incubation Program of Hebei University (Grant Nos. 521100223213, 521000981248, 521000981377 and 8012605), Hebei University (050001-521100302025, 050001-513300201004), the Scientific Research Foundation of Hebei Agricultural University (YJ201939), the National Natural Science Foundation of China (No. 51971245, 52022088), the Photoexcitonix Project at Hokkaido University. We thank the TEM technical support provided by the Microanalysis Center, College of Physics Science and Technology, Hebei University.

## Author contributions

Y.L. and J.Y. conceived the project and contributed to the design of the experiments and analysis of the data. X.B., D.Y., and C.Y., performed the catalysts preparation and characterizations. L.Z., Y.G., and X.S., carried out the in-situ TEM and HAADF-STEM characterizations. Y.L. and J.Y. wrote the paper. All the authors discussed the results and commented on the manuscript.

## Competing interests

The authors declare no competing interests.
