## [Peer Review File · Nature Communications]

Cu-based high-entropy two-dimensional oxide as stable and active photothermal catalystREVIEWER COMMENTS

Reviewer #1 (Remarks to the Author):

This is a great paper on an emerging topic on catalysis. The advantage of high entropy materials was cleverly demonstrated. It can be published after consideration of the following items:

- (1) The temperature used in their "templated" synthesis of high entropy materials is only 450 C. This is very low. Their XRD pattern is very broad, indicating very amorphous structures. I would suspect that the material is in a metastable state. I would like to the authors to comment on their crystallinity of their samples.
- (2) Cu-containing oxides are known for their CO oxidation catalytic activities. Can the authors test their catalysts for CO oxidation activities?
- (3) A recent related paper on Cu-containing surface high entropy fluorite catalyst (Defect Engineering of Ceria Nanocrystals for Enhanced Catalysis via a High-Entropy Oxide Strategy, ACS Cent. Sci. 2022, 8, 8, 1081–1090) should be cited.

Reviewer #2 (Remarks to the Author):

The work of Li et al. presents noteworthy it results.

The $\text{Cu}_2\text{Zn}_1\text{Al}_{0.5}\text{Ce}_5\text{Zr}_{0.5}\text{O}_x$ 2D catalyst converted 36%+ photochemical energy into chemical energy. The CO production rate was 4 fold what reported in the literature.

The originality of the work mainly lies in the composition of the 2D material, which, despite the inner complexity of the composition itself, seems to have been synthesized with quite widely available methods, which is positive and very significant to the field.

The work support the conclusions and claim. The materials' characterization has been thoroughly executed: the data and its interpretation is detailed and supported by different analytical approaches. I believe there is enough detail in the methods to be reproduced.

The DFT analysis backs up the experimental data. The DFT method is very well explained. So are the DFT data.

The methodology seems very sound, although I would suggest to add error bars and to mention very clear how data uncertainty was taken into account.

Said this, I have minor comments (which are also highlighted in the attachments - documents marked up). The numbers at the beginning of each sentence include the manuscript line (see also attachment).

1. 27. Revise language + unclear
2. 33. Revise language (suggestion provided)
3. 34. Revise language (suggestion provided)
4. 45. It would be more informative to mention what actual industrial applications are
5. 58. Consider using more informative wording as these adjectives sound vague and uninformative

6. 72. I suggest stating the novelty of the work more strongly
7. Ideally improve quality of several images (refer to marked manuscript) as they are mostly blurred
8. Throughout the manuscript, you may consider changing "hydrogenation performance" to "activity"
9. I suggest changing section's title (activity vs, performance)
10. 147. I would reword the sentence (see marked up manuscript)
11. 154 I would reword the sentence (see marked up manuscript)
12. Please, add error bars and uncertainty to all your graphs and data
13. Please, note that other minor comments to the language are included in the attachments

Reviewer #1 (Remarks to the Author):

This is a great paper on an emerging topic on catalysis. The advantage of high entropy materials was cleverly demonstrated. It can be published after consideration of the following items:

(1) The temperature used in their "templated" synthesis of high entropy materials is only 450 °C. This is very low. Their XRD pattern is very broad, indicating very amorphous structures. I would suspect that the material is in a metastable state. I would like to the authors to comment on their crystallinity of their samples.

Response: The authors thank for the reviewer's constructive comment. From our preparation experience, the crystallinity of 2D high entropy metal oxides largely depends on the type of metal element (alkaline earth metals (Mg, Ca) could enhance the crystallinity of 2D high entropy metal oxides), and is not significantly related to the annealing temperature. As shown in Supplementary Fig. 9, the pristine 2D $\text{Cu}_2\text{Zn}_1\text{Al}_{0.5}\text{Ce}_5\text{Zr}_{0.5}\text{O}_x$ and the 2D $\text{Cu}_2\text{Zn}_1\text{Al}_{0.5}\text{Ce}_5\text{Zr}_{0.5}\text{O}_x$ after 800 °C of RWGS have similar broaden XRD peaks. It indicates that the crystallinity of 2D $\text{Cu}_2\text{Zn}_1\text{Al}_{0.5}\text{Ce}_5\text{Zr}_{0.5}\text{O}_x$ has not clearly improved under high temperature treatment. In contrast, Supplementary Fig. 2 shows the relative narrow XRD peaks of 2D $\text{Ce}_1\text{Cu}_1\text{Mn}_1\text{Mg}_1\text{Al}_1\text{Co}_1\text{La}_1\text{Zr}_1\text{Ca}_1\text{Y}_1\text{O}_x$ after 450 °C of annealing process, depicting a high crystallinity.

Supplementary Fig. 9 | The XRD patterns of pristine 2D $\text{Cu}_2\text{Zn}_1\text{Al}_{0.5}\text{Ce}_5\text{Zr}_{0.5}\text{O}_x$ and 2D $\text{Cu}_2\text{Zn}_1\text{Al}_{0.5}\text{Ce}_5\text{Zr}_{0.5}\text{O}_x$ after 800 °C of RWGS.

Supplementary Fig. 2 | XRD pattern of 2D $\text{Ce}_1\text{Cu}_1\text{Mn}_1\text{Mg}_1\text{Al}_1\text{Co}_1\text{La}_1\text{Zr}_1\text{Ca}_1\text{Y}_1\text{O}_x$.

(2) Cu-containing oxides are known for their CO oxidation catalytic activities. Can the authors test their catalysts for CO oxidation activities?

Response: The authors thank for the reviewer’s comment, we have tested the CO oxidation activity of 2D $\text{Cu}_2\text{Zn}_1\text{Al}_{0.5}\text{Ce}_5\text{Zr}_{0.5}\text{O}_x$ as shown in Supplementary Fig. 14, and added the description in the revised manuscript.

“In addition, 2D $\text{Cu}_2\text{Zn}_1\text{Al}_{0.5}\text{Ce}_5\text{Zr}_{0.5}\text{O}_x$ also showed the activity of CO oxidation (Supplementary Fig. 14),¹ indicating the potential for catalytic versatility.”

Supplementary Fig. 14 | The CO oxidation performance of 2D $\text{Cu}_2\text{Zn}_1\text{Al}_{0.5}\text{Ce}_5\text{Zr}_{0.5}\text{O}_x$.

“The CO oxidation was tested by the fixed-bed reactor (XM190708-007, DALIAN ZHONGJIARUILIN LIQUID TECHNOLOGY CO., LTD) in continuous flow form. Typically, 15 mg of 2D $\text{Cu}_2\text{Zn}_1\text{Al}_{0.5}\text{Ce}_5\text{Zr}_{0.5}\text{O}_x$ was placed in a quartz flow reactor and the feeding gas of $\text{CO}/\text{O}_2/\text{Ar}/\text{N}_2 = 1/20/99/80$ with 100 sccm of flow rate was regulated by the mass flow controller. The reaction products were tested by gas chromatography (GC) 7890A equipped with FID and TCD detectors.”

(3) A recent related paper on Cu-containing surface high entropy fluorite catalyst

(Defect Engineering of Ceria Nanocrystals for Enhanced Catalysis via a High-Entropy Oxide Strategy, ACS Cent. Sci. 2022, 8, 8, 1081-1090) should be cited.

Response: Thanks for the reviewer's suggestion, we have cited this literature in the revised manuscript.

“In addition, 2D $\text{Cu}_2\text{Zn}_1\text{Al}_{0.5}\text{Ce}_5\text{Zr}_{0.5}\text{O}_x$ also showed the activity of CO oxidation (Supplementary Fig. 14),¹ indicating the potential for catalytic versatility.”

“56 Sun, Y. *et al.* Defect engineering of ceria nanocrystals for enhanced catalysis via a high-entropy oxide strategy. *ACS Cent. Sci.* **8**, 1081-1090, (2022).”

Reviewer #2 (Remarks to the Author):

The work of Li et al. presents noteworthy results.

The $\text{Cu}_2\text{Zn}_1\text{Al}_{0.5}\text{Ce}_5\text{Zr}_{0.5}\text{O}_x$ 2D catalyst converted 36%+ photochemical energy into chemical energy. The CO production rate was 4 fold what reported in the literature.

The originality of the work mainly lies in the composition of the 2D material, which, despite the inner complexity of the composition itself, seems to have been synthesized with quite widely available methods, which is positive and very significant to the field.

The work support the conclusions and claim. The materials' characterization has been thoroughly executed: the data and its interpretation is detailed and supported by different analytical approaches. I believe there is enough detail in the methods to be reproduced.

The DFT analysis backs up the experimental data. The DFT method is very well explained. So are the DFT data.

The methodology seems very sound, although I would suggest to add error bars and to mention very clear how data uncertainty was taken into account.

Said this, I have minor comments (which are also highlighted in the attachments-documents marked up). The numbers at the beginning of each sentence include the manuscript line (see also attachment).

Response: We are grateful for the reviewer's comprehensive review and suggestions. We have revised the whole article according to your comments and highlight in blue color in the revised manuscript.

1. 27. Revise language + unclear

Response: The authors thank for the reviewer's valuable suggestion, and we have revised this unclear sentence in the revised manuscript.

“and a PVP templated method is invented for generally synthesizing six-eleven dissimilar elements as high-entropy two-dimensional (2D) materials.”

2. 33. Revise language (suggestion provided)

3. 34. Revise language (suggestion provided)

Response: Thanks for the reviewer's advices, and we have revised the sentences in the revised manuscript according to the marks.

“it exhibited a record photochemical energy conversion efficiency of 36.2%, with a CO generation rate of 248.5 mmol g⁻¹ h⁻¹”

“and a PVP templated method could generally and large-scale synthesize high-entropy two-dimensional (2D) materials.”

4. 45. It would be more informative to mention what actual industrial applications are

Response: Thanks for the reviewer's good suggestion, we have added some industrial applications in the revised manuscript.

“Cu-based nanomaterials are the benchmark catalysts of diverse industrial processes, such as methanol steam reforming,² methanol synthesis,^{3,4} water gas shift reaction,⁵”

“11 Li, D. *et al.* Induced activation of the commercial Cu/ZnO/Al₂O₃ catalyst for the steam reforming of methanol. *Nat. Catal.* **5**, 99-108, (2022).

12 Laudenschleger, D., Ruland, H. & Muhler, M. Identifying the nature of the active sites in methanol synthesis over Cu/ZnO/Al₂O₃ catalysts. *Nat. Commun.* **11**, 3898, (2020).

13 Zhao, H. *et al.* The role of Cu₁-O₃ species in single-atom Cu/ZrO₂ catalyst for CO₂ hydrogenation. *Nat. Catal.* **5**, 818-831, (2022).

14 Shi, C. C. *et al.* Outdoor sunlight-driven scalable water-gas shift reaction through novel photothermal device-supported CuO_x/ZnO/Al₂O₃ nanosheets with a hydrogen generation rate of 192 mmol g⁻¹ h⁻¹. *J. Mater. Chem. A* **8**, 19467-19472, (2020).”

5. 58. Consider using more informative wording as these adjustives sound vague and uninformative

Response: Thanks for the reviewer's suggestion, and we have revised the sentence in the revised manuscript.

“Therefore, regulating the structure of Cu-based nanocatalysts to obtain high catalytic stability and activity at high temperatures is important for the catalytic science.”

6. 72. I suggest stating the novelty of the work more strongly

Response: Thanks for the reviewer's suggestion, and we added a sentence to emphasize our novelty of the work.

“This work offers a new pathway for low-temperature synthesizing high-entropy metal oxide nanocatalysts to realize the synergism of catalytic stability and activity of Cu based nanocatalysts.”

7. Ideally improve quality of several images (refer to marked manuscript) as they are mostly blurred

Response: Thanks for the reviewer's very valuable suggestion, we are sorry for low quality of images when the manuscript compressed to PDF format. Therefore, we have provided those figures with high resolution.

Fig. 1 The preparation and characterizations of high-entropy 2D materials. **a** The preparation diagram of 2D high-entropy materials. **b** The TEM image and Ce, Cu, Mn, Mg, Al, Co, La, Zr, Ca, Y, O elemental mapping images of 2D $\text{Ce}_1\text{Cu}_1\text{Mn}_1\text{Mg}_1\text{Al}_1\text{Co}_1\text{La}_1\text{Zr}_1\text{Ca}_1\text{Y}_1\text{O}_x$. **c** TEM image, **d** XRD pattern, **e** HAADF-STEM image of 2D $\text{Cu}_2\text{Zn}_1\text{Al}_{0.5}\text{Ce}_5\text{Zr}_{0.5}\text{O}_x$. **f** Cu, Zn, Al, Ce, Zr, O elemental mapping images of 2D $\text{Cu}_2\text{Zn}_1\text{Al}_{0.5}\text{Ce}_5\text{Zr}_{0.5}\text{O}_x$. The scale bars in **b**, **c**, **e**, **f** are 300 nm, 2 μm , 2 nm, 50 nm, respectively.

Fig. 2 Thermal RWGS performance of catalysts. **a** Thermal RWGS performance of 2D Cu₂Zn₁Al_{0.5}Ce₅Zr_{0.5}O_x, 2D Cu₂Ce₇O_x, commercial CuZnAlO_x (Cu₆Zn₃Al₁). **b** Visual contrast diagram of the RWGS CO production rates of 2D Cu₂Zn₁Al_{0.5}Ce₅Zr_{0.5}O_x (This work) and other advanced catalysts at 500 °C. **c** The RWGS stability of 2D Cu₂Zn₁Al_{0.5}Ce₅Zr_{0.5}O_x, 2D Cu₂Ce₇O_x and Cu₆Zn₃Al₁ under 450 °C. **d** The CO selectivity of 2D Cu₂Zn₁Al_{0.5}Ce₅Zr_{0.5}O_x for thermal RWGS at different temperature. The errors of 2D Cu₂Zn₁Al_{0.5}Ce₅Zr_{0.5}O_x show standard deviation.

Fig. 3 In-situ characterizations of catalysts. **a, b** In-situ TEM observations of the 2D $\text{Cu}_2\text{Zn}_1\text{Al}_{0.5}\text{Ce}_5\text{Zr}_{0.5}\text{O}_x$, 2D $\text{Cu}_2\text{Ce}_7\text{O}_x$ at different temperatures of RWGS. **c** H_2 -TPR curves of the 2D $\text{Cu}_2\text{Zn}_1\text{Al}_{0.5}\text{Ce}_5\text{Zr}_{0.5}\text{O}_x$ and 2D $\text{Cu}_2\text{Ce}_7\text{O}_x$. **d, e** The Cu 2p XPS spectra of 2D $\text{Cu}_2\text{Zn}_1\text{Al}_{0.5}\text{Ce}_5\text{Zr}_{0.5}\text{O}_x$ and 2D $\text{Cu}_2\text{Ce}_7\text{O}_x$ before and after the oxidation process. The scale bars in **a, b** are 1 μm .

Supplementary Fig. 18 | a-g The CO generation rate of solar heating RWGS through TiC-based solar heating device loaded with 100 g 2D Cu₂Zn₁Al_{0.5}Ce₅Zr_{0.5}O_x under ambient sunlight irradiation, on December 12, 13, 14, 17, 18, 20, 21, 2021, in Baoding City, China.

Supplementary Fig. 19 | a-g The outdoor sunlight intensity on December 12, 13, 14, 17, 18, 20, 21, 2021, in Baoding City, China.

8. Throughout the manuscript, you may consider changing "hydrogenation performance" to "activity"

Response: Thanks for the reviewer's good suggestion, we have changed word "performance" to "activity" in the abstract.

"but also improved its CO₂ hydrogenation activity to a pure CO production rate of 417.2

mmol g⁻¹ h⁻¹ at 500 °C,”

9. I suggest changing section's title (activity vs, performance)

Response: Thanks for the reviewer's good suggestion, we have changed word “performance” to “activity” in section's title.

“The CO₂ hydrogenation activity”

10. 147. I would reword the sentence (see marked up manuscript)

Response: The authors thank for the reviewer's very valuable suggestion. Therefore, we changed the sentence according suggestion.

“Fig. 2c displays the thermal RWGS stability of 2D Cu₂Zn₁Al_{0.5}Ce₅Zr_{0.5}O_x at 450 °C for 72 hours.”

11. 154 I would reword the sentence (see marked up manuscript)

Response: The authors thank for the reviewer's advice, and we have amended this wrong sentence.

“Additionally, the 2D Cu₂Zn₁Al_{0.5}Ce₅Zr_{0.5}O_x showed 100 % selectivity for CO₂ converted as CO (Fig. 2d).”

12. Please, add error bars and uncertainty

Response: The authors thank for the reviewer's valuable suggestion, so we have added error bars in Fig. 2a and Fig 5a. Note that, the photothermal RWGS demonstration test (Fig. 5c, d) need a large amount of catalysts and feeding gas. Therefore, we only test once and there were no additional error bars in Fig. 5c, d.

Fig. 2 Thermal RWGS performance of catalysts. **a** Thermal RWGS performance of 2D Cu₂Zn₁Al_{0.5}Ce₅Zr_{0.5}O_x, 2D Cu₂Ce₇O_x, commercial CuZnAlO_x (Cu₆Zn₃Al₁). **b** Visual contrast diagram of the RWGS CO production rates of 2D Cu₂Zn₁Al_{0.5}Ce₅Zr_{0.5}O_x (This work) and other advanced catalysts at 500 °C. **c** The RWGS stability of 2D Cu₂Zn₁Al_{0.5}Ce₅Zr_{0.5}O_x, 2D Cu₂Ce₇O_x and Cu₆Zn₃Al₁ under 450 °C. **d** The CO selectivity of 2D Cu₂Zn₁Al_{0.5}Ce₅Zr_{0.5}O_x for thermal RWGS at different temperature. The errors of 2D Cu₂Zn₁Al_{0.5}Ce₅Zr_{0.5}O_x show standard deviation.

Fig. 5 The photothermal RWGS performance of 2D $\text{Cu}_2\text{Zn}_1\text{Al}_{0.5}\text{Ce}_5\text{Zr}_{0.5}\text{O}_x$. **a** The temperature of 2D $\text{Cu}_2\text{Zn}_1\text{Al}_{0.5}\text{Ce}_5\text{Zr}_{0.5}\text{O}_x$ and CO production rate of photothermal RWGS through 2D $\text{Cu}_2\text{Zn}_1\text{Al}_{0.5}\text{Ce}_5\text{Zr}_{0.5}\text{O}_x$ under different sunlight irradiation. **b** The CO generation rate of photothermal RWGS through 2D $\text{Cu}_2\text{Zn}_1\text{Al}_{0.5}\text{Ce}_5\text{Zr}_{0.5}\text{O}_x$ under 2 suns irradiation and light off working conditions. **c** The CO generation rate of photothermal RWGS demonstration with 100 g of 2D $\text{Cu}_2\text{Zn}_1\text{Al}_{0.5}\text{Ce}_5\text{Zr}_{0.5}\text{O}_x$ under different sunlight irradiation. **d** The STC efficiency of photothermal RWGS demonstration with 100 g of 2D $\text{Cu}_2\text{Zn}_1\text{Al}_{0.5}\text{Ce}_5\text{Zr}_{0.5}\text{O}_x$ under different sunlight irradiation. **e** The photograph of photothermal RWGS demonstration equipped with reflector in Hebei University. **f** The CO yield of photothermal RWGS demonstration equipped with reflector under outdoor sunlight irradiation, on December 12, 13, 14, 17, 18, 20, 21, 2021, in Baoding City, China. The errors show standard deviation.

13. Please, note that other minor comments to the language are included in the attachments

Response: Thanks for the reviewer's good suggestion, we have corrected unsuitable word according to the attachments.

“and a PVP templated method is invented for generally synthesizing six-eleven dissimilar elements as high-entropy two-dimensional (2D) materials.”

“The freeze-drying process was applied to obtain solids of 2D PVP micelles loaded with various metal ions.”

“which is the fingerprint feature of high-entropy materials.”

“Meanwhile, the instruments, chemicals, and steps required for this PVP templated method are simple and inexpensive.”

“Fig. 2a shows the RWGS CO production...”

“It confirmed that using metal oxides such as CeO₂ as support to introduce SMSI can weaken the sintering of Cu species.”

References

- 1 Sun, Y. *et al.* Defect Engineering of Ceria Nanocrystals for Enhanced Catalysis via a High-Entropy Oxide Strategy. *ACS Cen. Sci.* **8**, 1081-1090, (2022).
- 2 Li, D. *et al.* Induced activation of the commercial Cu/ZnO/Al₂O₃ catalyst for the steam reforming of methanol. *Nat. Catal.* **5**, 99-108, (2022).
- 3 Laudenschleger, D., Ruland, H. & Muhler, M. Identifying the nature of the active sites in methanol synthesis over Cu/ZnO/Al₂O₃ catalysts. *Nat. Commun.* **11**, 3898, (2020).
- 4 Zhao, H. *et al.* The role of Cu₁-O₃ species in single-atom Cu/ZrO₂ catalyst for CO₂ hydrogenation. *Nat. Catal.* **5**, 818-831, (2022).
- 5 Shi, C. C. *et al.* Outdoor sunlight-driven scalable water-gas shift reaction through novel photothermal device-supported CuO_x/ZnO/Al₂O₃ nanosheets with a hydrogen generation rate of 192 mmol g⁻¹ h⁻¹. *J. Mater. Chem. A* **8**, 19467-19472, (2020).

REVIEWERS' COMMENTS

Reviewer #1 (Remarks to the Author):

The authors have addressed the issues. It can be accepted.

Reviewer #2 (Remarks to the Author):

The authors have well addressed all the comments raised and provided a complete explanation. They have revised all the figures in the manuscript to a higher standard. The manuscript can now be accepted.

Reviewer #1 (Remarks to the Author):

The authors have addressed the issues. It can be accepted.

Response: The authors thank for the reviewer's constructive comment for this manuscript.

Reviewer #2 (Remarks to the Author):

The authors have well addressed all the comments raised and provided a complete explanation. They have revised all the figures in the manuscript to a higher standard. The manuscript can now be accepted.

Response: We greatly appreciate the guidance of reviewer to improve the quality of this manuscript.